# Assessing the Potential of Enhanced Resolution Gridded Passive Microwave Brightness Temperatures for Retrieval of Sea Ice Parameters

**Walter N. Meier ***  **and J. Scott Stewart**

National Snow and Ice Data Center, Cooperative Institute for Research in Environmental Sciences, University of Colorado, Boulder, CO 80309, USA; scotts@colorado.edu
*   Correspondence: walt@nsidc.org; Tel.: +1-303-735-6276

**Abstract:** A new enhanced resolution gridded passive microwave brightness temperature (TB) product is used to estimate sea ice concentration and motion. The effective resolution of the TBs is found to be roughly twice that of the standard 25 km resolution, though the gridded resolution of the distributed product is higher. Enhanced resolution sea ice concentrations from the Bootstrap algorithm show more detail in the sea ice, including relatively small open water regions within the ice pack. Sea ice motion estimates from the enhanced resolution TBs using a maximum cross-correlation method show a smoother motion circulation pattern; in comparison to buoys, RMS errors are 15–20% lower than motion estimates from the standard resolution fields and the magnitude of the bias is smaller as well. The enhanced resolution product includes other potentially beneficial characteristics, including twice-daily grids based on local time of day and a complete timeseries of data from nearly all multi-channel passive microwave radiometers since 1978. These enhanced resolution TBs are potential new source for long-term records of sea ice concentration, motion, age, melt, as well as salinity and ocean-atmosphere fluxes.

**Keywords:** sea ice; passive microwave; remote sensing; Arctic

---

## 1. Introduction

Passive microwave remote sensing data have been a key source of sea ice information, providing a >40-year time series of sea ice conditions, e.g., [1,2]. The record of the significantly declining Arctic sea ice extent (e.g., [3]) is an iconic indicator of global climate change. Passive microwave (PM) imagery is particularly useful for the polar regions for several reasons. First, PM emission from the sea ice surface does not require sunlight and is generally not affected by the atmosphere (e.g., clouds)—i.e., it provides all-sky coverage. Second, PM sensors have a wide swath that completely covers sea ice regions at daily intervals. Finally, there has been at least one multi-sensor satellite-borne PM sensor in operation since October 1978, providing a continuous long-term time series of daily sea ice conditions (with every other day coverage for 1978 to August 1987). The sensors have been largely consistent in frequency and antenna size with at least some overlap when two or more sensors were operating simultaneously; this allows precise intercalibration to assure consistency in the long-term record.

There are, however, several limitations of PM sensors for sea ice, including reduced effectiveness when there is surface melt and other surface complexities (thin ice, characteristics of overlying snow cover) (e.g., [4]). The most significant limitation of passive microwave sensors is their low spatial resolution (as low as on the order of ~50–75 km). This limits the detail that can be obtained by PM sensors and contributes to large uncertainties under certain conditions, such as near the ice edge and in coastal regions.

Recently, a new PM brightness temperature (TB) product has been introduced that provides enhanced spatial resolution TB fields on the Equal-Area Scalable Earth Grid 2.0 (EASE2) [5]. The product uses overlapping footprints to synthesize a higher resolution. This product is potentially a source of improved sea ice parameters, including concentration, extent, and motion. Here we explore the potential of these fields for such sea ice estimates.

## 2. Materials and Methods

The first multi-channel PM sensor was the NASA Scanning Multichannel Microwave Radiometer (SMMR) on the Nimbus-7 platform, launched in October 1978. This was followed by a series of Special Sensor Microwave Imager (SSMI) and Special Sensor Microwave Imager/Sounder (SSMIS) instruments on U.S. Department of Defense Meteorological Satellite Program (DSMP) "F-series" platforms. Currently (as of 31 July 2020), three SSMIS sensors are operating on the DMSP F-16, F-17, and F-18 platforms.

More recently, a newer set of sensors have been launched: first, the Advanced Microwave Scanning Radiometer (AMSR-E) on NASA's Earth Observing System (EOS) Aqua platform operating from 2002 to 2011. AMSR-E has been followed by the JAXA AMSR2 on the Global Change Observation Mission-Water (GCOM-W) Shizuku platform, which was launched in May 2012 and continues to operate (as of 31 July 2020). The AMSR sensors have a larger antenna than the SMMR-SSMI-SSMIS instruments and thus higher spatial resolution. However, the different resolution yields less consistency in the sea ice timeseries [6].

Sea ice concentration has been derived from PM TBs via empirically-derived algorithms, based on the signature for pure surface types (i.e., 100% sea ice and 100% open water). Several algorithms have been developed over the years. The algorithms generally differ in their selection of microwave channels (frequency and polarization) employed and in the implementation details. Typically, the algorithms use a combination of channels from among horizontal (H) and vertical (V) polarization of ~19 GHz and ~37 GHz frequencies, though some use higher frequency (near-90 GHz) channels [7,8]. Two algorithms with a long heritage are the NASA Team [9] and Bootstrap [6], both developed at the NASA Goddard Space Flight Center. Sea ice concentration fields from these two algorithms are distributed by the NASA Snow and Ice Distributed Active Archive Center (DAAC) at the National Snow and Ice Data Center (NSIDC) [10,11].

The NASA Team algorithm uses a ratio of polarization differences (19H and 19V), the "polarization ratio", and a ratio of frequency difference (19V and 37V), the "gradient ratio". The Bootstrap algorithm uses two relationships: between 37H and 37V for high concentration regions, and between 19V and 37V for lower concentration regions.

These two algorithm products use daily gridded polar stereographic TBs from the NSIDC DAAC as their source [12]. The gridded TBs are produced at nominal 25 km (12.5 km for near-90 GHz) resolution from swath TBs produced by Remote Sensing Systems, Inc. [13] using a simple drop-in-the-bucket method, where the daily TB value at each grid cell is the average of any sensor footprints whose center falls within the grid cell in the 24-h period. Thus, while the gridded resolution is 25 km for the sea ice algorithm channels, the effective spatial resolution is limited by the sensor footprint resolution and is coarser than the gridded resolution. In addition, the daily average represents a temporal "smearing" by averaging footprints from different times within a 24-h period.

Sea ice motion is another parameter commonly derived from passive microwave TBs. Motion can be derived by feature-matching TB signatures from two spatially coincident images separated by a period of time. One such approach that has been commonly implemented is the Maximum Cross-Correlation (MCC) method [14], which calculates the correlation of a feature (grid cell) in the first image with features (grid cells) within a "window" region surrounding the original grid cell, using a sliding "search box". A correlation is calculated for all features within the window and the correlation peak is deemed to be the new location of the feature in the second image. The distance between the features is found and dividing by the time separation yields a motion velocity. The MCC is used for

PM sources of the NSIDC DAAC sea ice motion product [15,16]. The advantage of passive microwave imagery is its all-sky capability and nearly-complete daily coverage. However, a major limitation of PM for motion estimates is the coarse spatial resolution. With gridded data, the resolution of the feature tracking is the same as the grid resolution—i.e., motions are obtained only in integral grid cells. For the NSIDC product, an oversampling method is applied, shifting the search box fractions of a grid cell and finding the correlation for each fractional shift of the search box. A 4X oversampling yields a motion discretization of 0.25 grid cells or 6.25 km for the 25 km grid. Another limitation is the use of daily composite TBs, which means the time separation is assumed to be 24 h. However, the time separation may be as low as ~2 h and as high as ~48 h. Furthermore, the averaging of multiple passes over a day means there is a "temporal smearing", with the signatures from several passes averaged together.

The Calibrated Enhanced-Resolution Passive Microwave Daily EASE-Grid 2.0 Brightness Temperature (CETB) Earth Science Data Record product [5], distributed by the NSIDC DAAC, represents a substantial advance in gridded brightness temperatures. The CETB is produced from calibrated swath TBs from the Colorado State University Fundamental Climate Data Records on the EASE-Grid 2.0 (EASE2) [17]. For EASE2, all grid cells have equal area, in contrast to polar stereographic grids for which grid cell areas depends on latitude. As for the polar stereographic grids, the CETB includes standard drop-in-the-bucket average fields (GRD), but twice per day instead of a single daily average. For polar (Northern Hemisphere and Southern Hemisphere) grids, instead of the more common approach of compositing ascending and descending passes, a local-time-of-day (LTOD) approach is used. Each sensor footprint is binned into morning or evening composites based on LTOD.

The primary advancement of the CETB is the implementation of enhanced-resolution TB fields via the radiometer version of the Scatterometer Image Reconstruction (rSIR) method. Here we provide a brief summary of the approach; details are provided in [18,19]. The rSIR was originally developed for scatterometer data [20] and then adapted for passive microwave radiometers. It is a signal processing approach that uses knowledge of the measurement response function (MRF) of the sensor to synthesize the radiometric signal at a higher resolution from multiple overlapping sensor footprints (Figure 1). The gridded resolution of the enhanced field is determined by the signal-to-noise ratio, which is dependent on the input of the native sensor footprint resolution. For the SMMR-SSMI-SSMIS sensors, the rSIR method produces gridded fields at 6.25 km resolution for 19 GHz channels and 3.125 km for 37 GHz fields (Table 1). However, the effective resolution—i.e., the resolution of the smallest resolvable feature—is lower than the gridded resolution [17]; this is discussed further below.

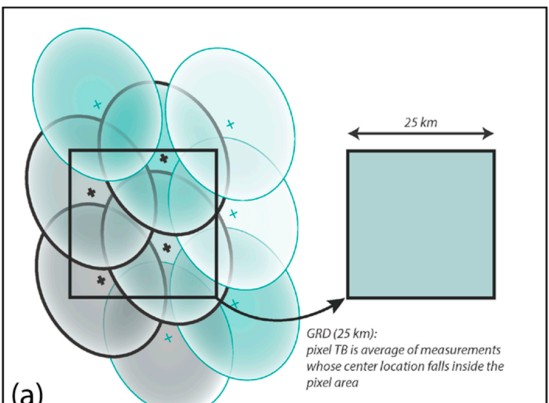 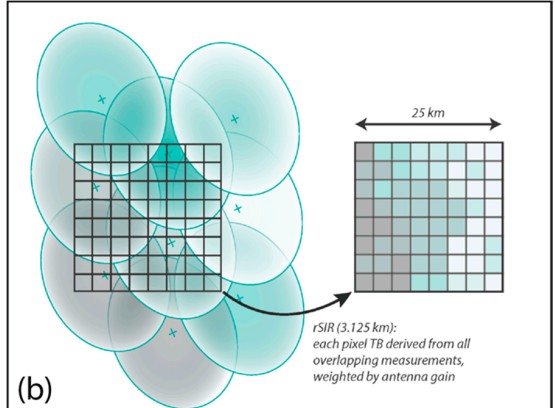

**Figure 1.** Schematics of (**a**) drop-in-the-bucket (GRD) method where all footprints whose centers are within the grid cell are averaged, and (**b**) rSIR where the sensor footprints are weighted based on the measurement response function to obtain enhanced resolution. Figure adapted from the Calibrated Enhanced-Resolution Passive Microwave Daily EASE-Grid 2.0 Brightness Temperature (CETB) product user's guide [21].

**Table 1.** Special Sensor Microwave Imager/Sounder (SSMIS) sensor instantaneous field of view (IFOV) (aka sensor footprint) and EASE2 grid spatial resolution for drop-in-the-bucket GRD and the radiometer Scatterometer Image Reconstruction (rSIR) fields.

| Frequency | Sensor IFOV (km) | GRD | rSIR |
|---|---|---|---|
| 19 GHz | $72 \times 44$ | 25.0 | 6.25 |
| 22 GHz | $72 \times 44$ | 25.0 | 6.25 |
| 37 GHz | $44 \times 26$ | 25.0 | 3.125 |
| 91 GHz | $15 \times 9$ | 12.5 | 3.125 |

The enhanced resolution fields are produced for morning and evening grids based on LTOD, as with the drop-in-the-bucket GRD fields. CETB rSIR and GRD fields are available for SMMR, all SSMI and SSMIS, and AMSR-E. Currently, the product is updated about once per year; a near-real-time product is under consideration. (There is also a related product for the NASA Soil Moisture Active Passive (SMAP) sensor on a slightly different EASE2 grid [22].)

Here we investigate the potential utility of the CETB for sea ice fields in two ways. First, we apply the Bootstrap algorithm Version 3 [23] to the CETB TB fields from the F17 SSMIS sensor to derive sea ice concentration. We compare sea ice concentration from the enhanced resolution rSIR TBs with concentration from the native 25 km GRD TBs. We also compare the rSIR approach with a bi-linear approach; this approach simply resamples the 25 km information to 3.125 km (or 6.25 km) via bi-linear interpolation. This provides a baseline to assess the relative advantage of rSIR. Next, we investigate the TB characteristics across a sharp boundary (where the TB values substantially change in a step-like manner, such as a coastline) to estimate an effective resolution. Then, we propose an "upscaled" approach to bin the high resolution rSIR grids into grids that are consistent with the effective resolution. In this paper, we focus on a case study in the Arctic region, but the CETB includes fields for the Antarctic. The goal of this paper is not to conduct a full-scale validation but rather to illustrate the potential of the CETB for improved sea ice fields. In the future, we plan to investigate the CETB further, including examining performance in the Antarctic.

Second, we apply the CETB fields to retrieve sea ice motion estimates using the MCC method. We compare the upscaled rSIR estimates with the original GRD source fields. Motions estimated from International Arctic Buoy Programme (IABP) [24], buoy positions (included in the NSIDC product) are used for evaluation of the motions.

## 3. Results

### 3.1. Sea Ice Concentration Case Study

Here we present a case study of sea ice concentration for the Arctic on 1 March 2014 to evaluate the effect of enhanced resolution on the concentration retrievals. Version 3 of the Bootstrap algorithm was run largely as for standard SSMIS products [11]. The exception was the land-spillover correction to remove false ice that occurs from mixed water-land grid cells along the coast. These mixed grid cells can have a microwave signature that is interpreted by the algorithm as sea ice and results in erroneous sea ice detection along coastlines. The standard Bootstrap land-spillover method was developed for 25 km grids by comparing coastal grid cells with neighboring cells; here we generalized the approach to be applicable for any spatial resolution using a dilation and contraction procedure along the coastline.

We also developed a custom-created landmask because we found small errors in areas of the NSIDC EASE2 Northern Hemisphere landmask [25] in polar regions, where grid cells containing a mixture of primarily land ice and bare land are misclassified as open water. We created new EASE2 landmasks at all resolutions (3.125 km, 6.25 km, 12.5 km, and 25 km) based on the NASA Boston University (BU) Moderate resolution Imaging Spectroradiometer (MODIS) 500 m land classification product MC12Q1 [26] to obtain accurate sea ice fields in coastal areas (see Supplementary Materials).

Concentration fields were initially produced at two resolutions, the drop-in-the-bucket 25 km GRD field and on a 3.125 km rSIR grid. The Bootstrap algorithm employs both the 37 GHz (3.125 km grid for rSIR) and the 19 GHz (6.25 km grid for rSIR) channels. In order to have a consistent grid, the 19 GHz and 37 GHz rSIR TB resolutions were harmonized before being input into the algorithm. We examined both upscaling the 37 GHz to 6.25 km by binning and downscaling the 19 GHz to 3.125 km via bi-linear interpolation. Here we present the 19 GHz downscaling to 3.125 km to examine the highest potential resolution of the concentration fields. The 6.25 km fields (not shown) show similar characteristics.

The 1 March 2014 fields for the full Arctic are overall quite similar (Figure 2), indicating that the algorithm works as expected with the enhanced resolution TBs. To compare the fields in more detail, we next focus on a subset in the Bering Sea/Strait region (Figure 3). Here the differences between the two fields are more apparent. The "pixelation" of the 25 km GRD is apparent (Figure 3a), while the 3.125 km fields show much finer discretization (Figure 3c). Of particular note, is an open water region off the southeast Alaskan coast, which is visible within the 3.125 km rSIR grid, but is ice-covered within the 25 km GRD grid.

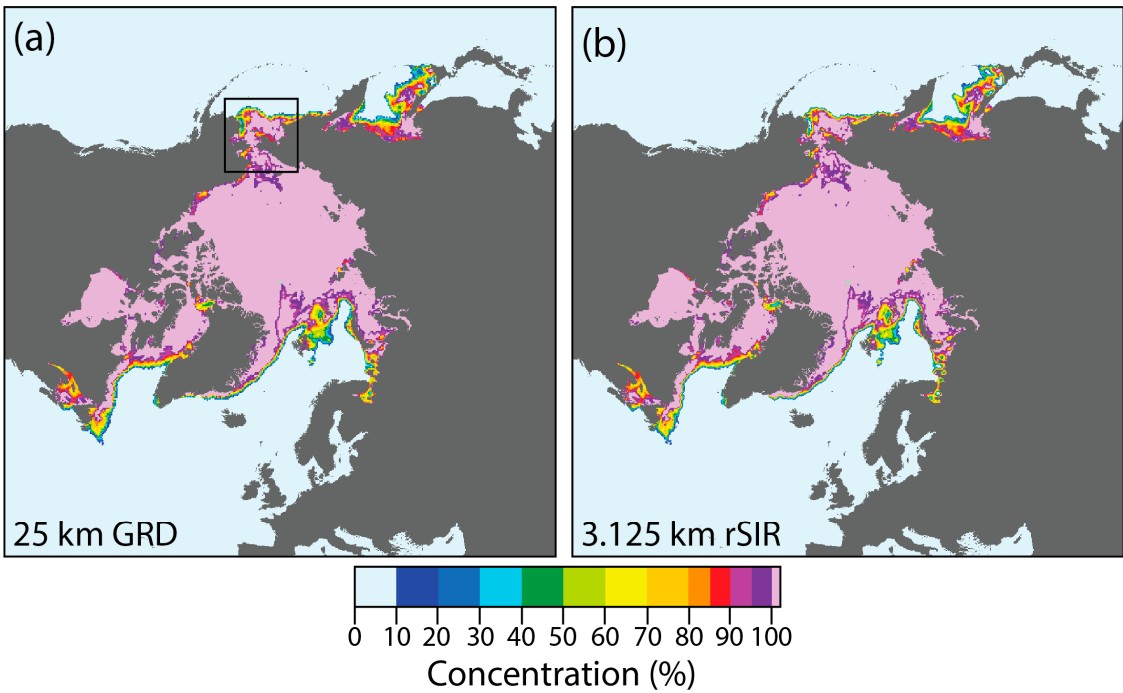

**Figure 2.** Northern Hemisphere sea ice concentration field from the Bootstrap algorithm for 1 March 2014 from (**a**) 25 km GRD TBs, and (**b**) 3.125 km rSIR brightness temperatures (TBs).

We also created a "basic" enhanced resolution field, also at 3.125 km, by simply bi-linearly interpolating the 25 km GRD TB fields to 3.125 km (see Supplementary Materials). This is a naïve method that serves as a baseline for the rSIR. If the rSIR is effective, it should provide greater detail than the bi-linear GRD field. Examining the differences between the enhanced resolution and the 25 km GRD, we see that this is indeed the case (Figure 3b,d). The bi-linear difference field (Figure 3b) shows a simple structure, with each grid cell showing a linear pattern in the concentration differences. Because it is a simple spatial interpolation, the open water feature is not present. In contrast, the rSIR field shows much more structure in the concentration differences (Figure 3d). In the open water region, the rSIR shows how the lower resolution 25 km GRD "smears" out the signal of the open water, spreading it out into neighboring grid cells so that in the open water area, rSIR shows lower concentration than GRD, while directly adjacent to the right (in Figure 3d), the rSIR has a higher concentration than GRD.

We confirm that the open water is a real feature via comparison with two fields (Figure 4). First, we visually inspect MODIS true color composite imagery from NASA WorldView [27]. Second, we compare

to the NSIDC Multisensor Analyzed Sea ice Extent (MASIE) [28]. MASIE is a reformatted version of the daily Interactive Multisensor Snow and Ice Mapping System (IMS) [29]. MASIE/IMS provides daily ice extent estimates at a 4 km resolution, using a 40% threshold, based on manual interpretation and integration of imagery by trained ice analysts. Several imagery sources are used, including visible/infrared, SAR, and a scatterometer. PM imagery from AMSR2 is used for MASIE analyses as well, generally as a last resort when no other data is available. The MASIE fields represent the best estimate of the ice edge location on a given day based on the available imagery and the judgment of expert ice analysts.

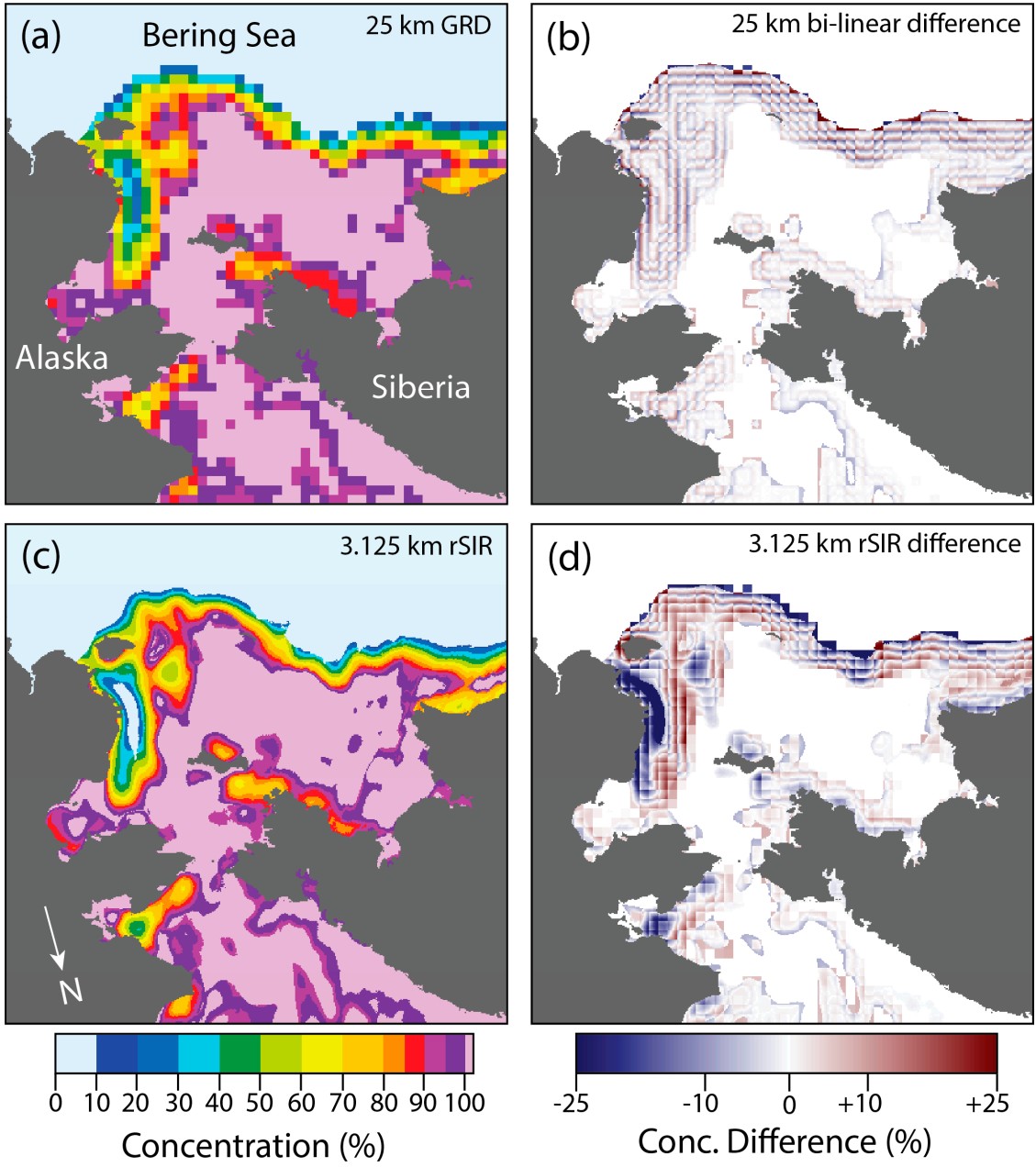

**Figure 3.** Bering Sea 1 March 2014 sea ice (**a**) 25 km GRD concentration, (**b**) of 3.125 km bi-linearly interpolated GRD minus 25 km GRD concentration difference, (**c**) 3.125 km rSIR concentration, and (**d**) 3.125 km rSIR minus 25 km GRD concentration difference.

The MASIE analysis maps the open water region and the rSIR field location matches it almost exactly. While clouds partially obscure the scene, MODIS clearly shows a dark region indicative of open water. MODIS actually appears to show a larger open water region with dark pixels beyond

the MASIE ice edge outline; however, upon closer inspection (see Supplementary Materials) the area outside of the MASIE outline is actually covered by thin ice, which looks dark gray, just slightly lighter than the water.

Thus, the rSIR fields appear to be able to capture finer-scale details than can be seen in the 25 km GRD fields. However, the rSIR field does not show fine-scale granularity at 3.125 km resolution that one expects for sea ice, especially in a highly variable region like the Bering Sea. Rather, the concentrations have a smoothly varying quality. In the concentration difference field (Figure 3d), it is clear that the rSIR is effectively fitting a smooth function to the TBs (and resulting concentrations). As noted above, there is much more structure in the rSIR compared to the simple GRD bi-linear field. So, there is an improvement, but not to the 3.125 km resolution of the grid. This is not surprising since as noted above, the rSIR output resolution is based on the signal processing methods of rSIR and the effective resolution—i.e., the minimum resolvable feature—is lower.

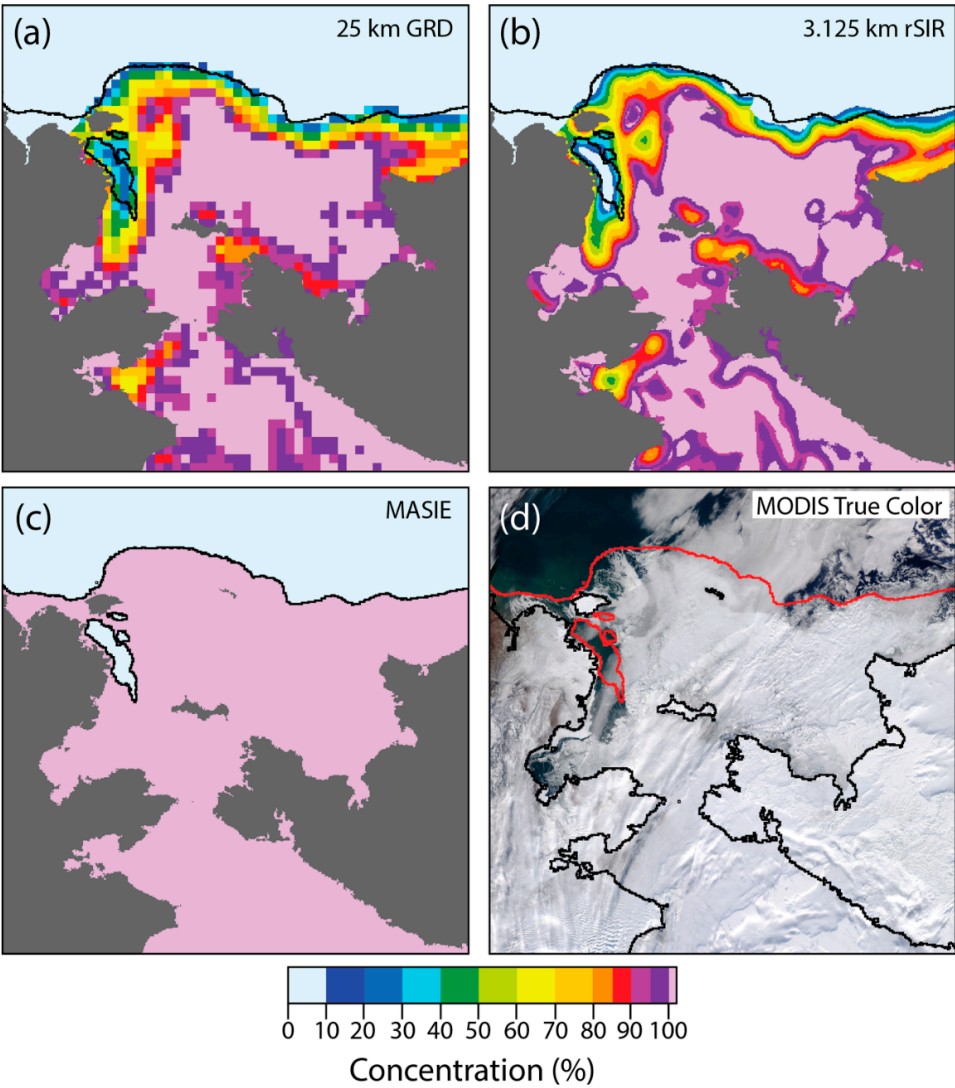

**Figure 4.** Bering Sea 1 March 2014 (**a**) 25 km GRD sea ice concentration, (**b**) 3.125 km rSIR concentration, (**c**) Multisensor Analyzed Sea ice Extent (MASIE) sea ice extent, (**d**) MODIS true color image. The MASIE ice edge is outlined in black (a–c) or red (d).

### 3.2. Effective Spatial Resolution of CETB

For sea ice concentration, there is no benefit in producing a field at a higher resolution than the input TB imagery can resolve. Thus, we next explore the effective resolution of the rSIR by examining

the variability of the TB fields across a sharp boundary. Here, we used the Kola Peninsula coastline in northern Finnoscandia during summer. This provides an almost straight horizontal (on the EASE2 grid) coastline. We track the TB values along a vertical transect of grid cells perpendicular to the coast from land to ocean (Figure 5).

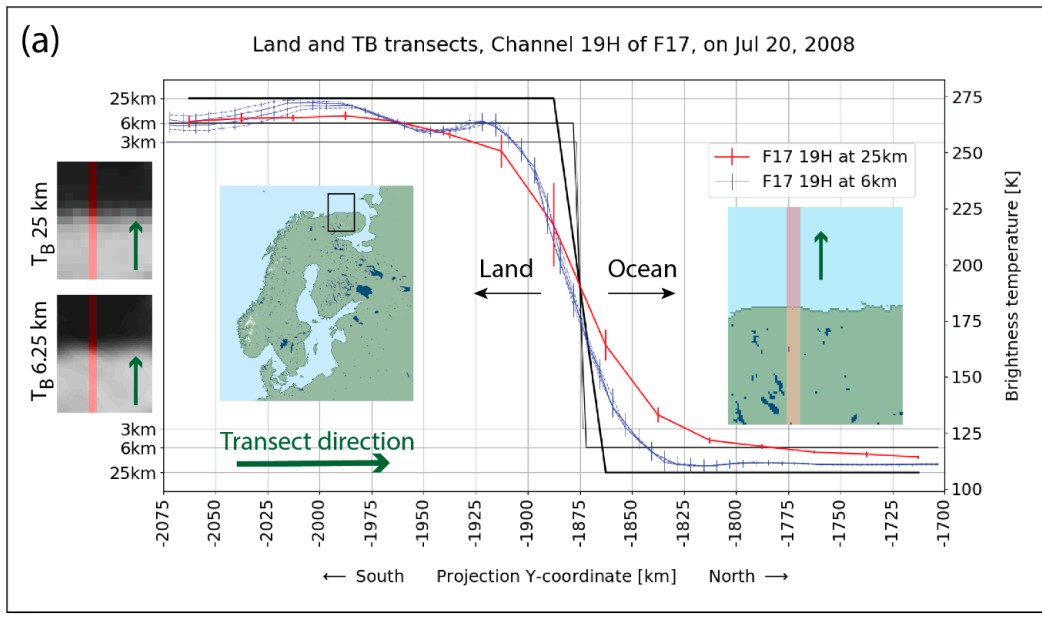

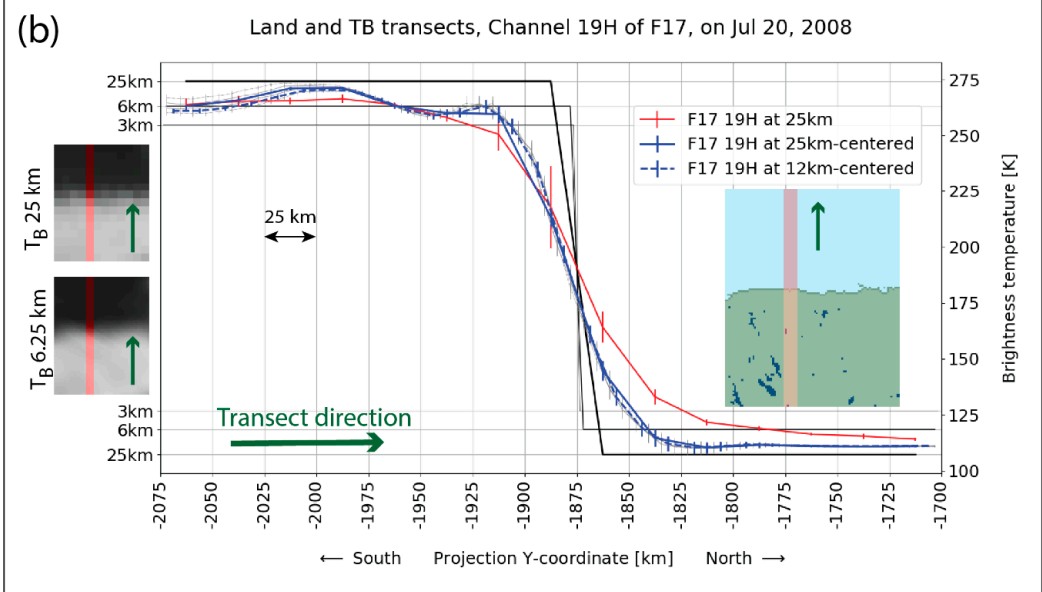

**Figure 5.** Transect across the land-ocean boundary, 20 July 2008. The main figure shows (**a**) the 19H TB (right y-axis) values for 25 km GRD (red) and 6.25 km rSIR (blue) and (**b**) the 19H TB values for 25 km GRD and 12.5 km and 25 km upscaled rSIR. The direction of the transect in the main figure is left (land) to right (ocean). Vertical gray lines indicate 25 km grid cells with data plotted at the center of each cell. The land-ocean boundary is at −1875 Y-coordinate. The insets on the left are TB images of the region and the insets within the main figure show the general Finnoscandian region and a close-up of the land with the overlaid transect (in pink).

Comparing the 25 km GRD 19H TB transect with the 6.25 km rSIR transect (Figure 5a) allows us to assess the length scale over which the signal reflects mixed land-ocean contribution. Over land (far left of Figure 5), the 19H TB values are ~270 K, typical for summer land, and open water (far right

of Figure 5) values are ~110 K. In both regions, the TB variability (vertical error bars) are low. Well before the coast (middle of figure) is reached, the land TB values start to decline and the decline in TBs continues well past the coast into the ocean; across the transition zone the TB variability increases substantially, indicating a mixing of signals from land and ocean. Overall, the transition from land to ocean in the 25 km GRD is ~150 km. If the true resolution were 25 km, we would expect a transition zone of no more than 25–50 km. However, while gridded at 25 km, the SSMIS 19 GHz sensor field of view (footprint) is 72 km × 44 km (Table 1). This explains the ~150 km transition length scale (70 to 75 km on each side of the coast corresponding to the large dimension of the footprint).

The transition region of the rSIR transect (Figure 5a) shows a shorter transition length of ~75 km. The rSIR fields also show lower variability (error bars on transect lines) than the GRD, indicating less mixing of ocean and land signal. Similar characteristics are seen for 37 GHz (see Supplementary Materials) but the transition length is shorter because of the smaller sensor footprint of the 37 GHz channels.

Since the rSIR effective resolution appears to be roughly twice that of the GRD, we upscaled the 3.125 km 37 GHz and 6.25 km 19 GHz TB grids to 12.5 km. We also created an upscaled rSIR 25 km grid to allow comparison between 25 km GRD and the same resolution from upscaled rSIR. The simplest approach would be to simply average all of the higher resolution grid cells within each 25 km grid cell. Another approach is to use only most central higher resolution grid cells within each lower resolution grid cell; for example, using only the central 2 × 2 3.125 km cells within the 8 × 8 3.125 km cells nested in a 25 km cell. This has the effect of capturing the most weight nearest the center of the 12.5 km or 25 km and minimizes any TB signal from outside the lower resolution cell.

Here we present the "centered" approach, though differences with the "binned" approach were found to be small. For the same Bering Sea region case study on 1 March 2014, both of the upscaled concentration grids, 25 km and 12.5 km, show noticeable improvement over the 25 km GRD field (Figure 6). In particular, both show the open water region off the southwest coast of Alaska. Thus, even at 25 km, the upscaled rSIR is able to observe more detail in the ice cover. Furthermore, both show a structure in the sea ice cover more consistent with the gridded resolution. The difference fields do not show as much of the smooth "fitting" apparent in the 3.125 km grid.

### 3.3. Sea Ice Motion from the CEBT

Sea ice motions were derived from 37H TBs using three different implementations: the nominal 25 km GRD grid, a 12.5 km bi-linearly interpolated GRD grid, and a 12.5 km rSIR grid. The MCC was run on LTOD Morning grids. An example vector field from 22 January 2014 illustrates differences between the fields (Figure 7). The 25 km GRD field is sparser; because of the coarser resolution, fewer vectors are retrieved. The vector field also has a "blocky" structure, with discontinuities between adjoining vectors due to the lower spatial resolution. The 12.5 km bi-linearly interpolated GRD has many more vectors but is also quite noisy. The rSIR field has the most vectors and has a smoother field that is likely more realistic.

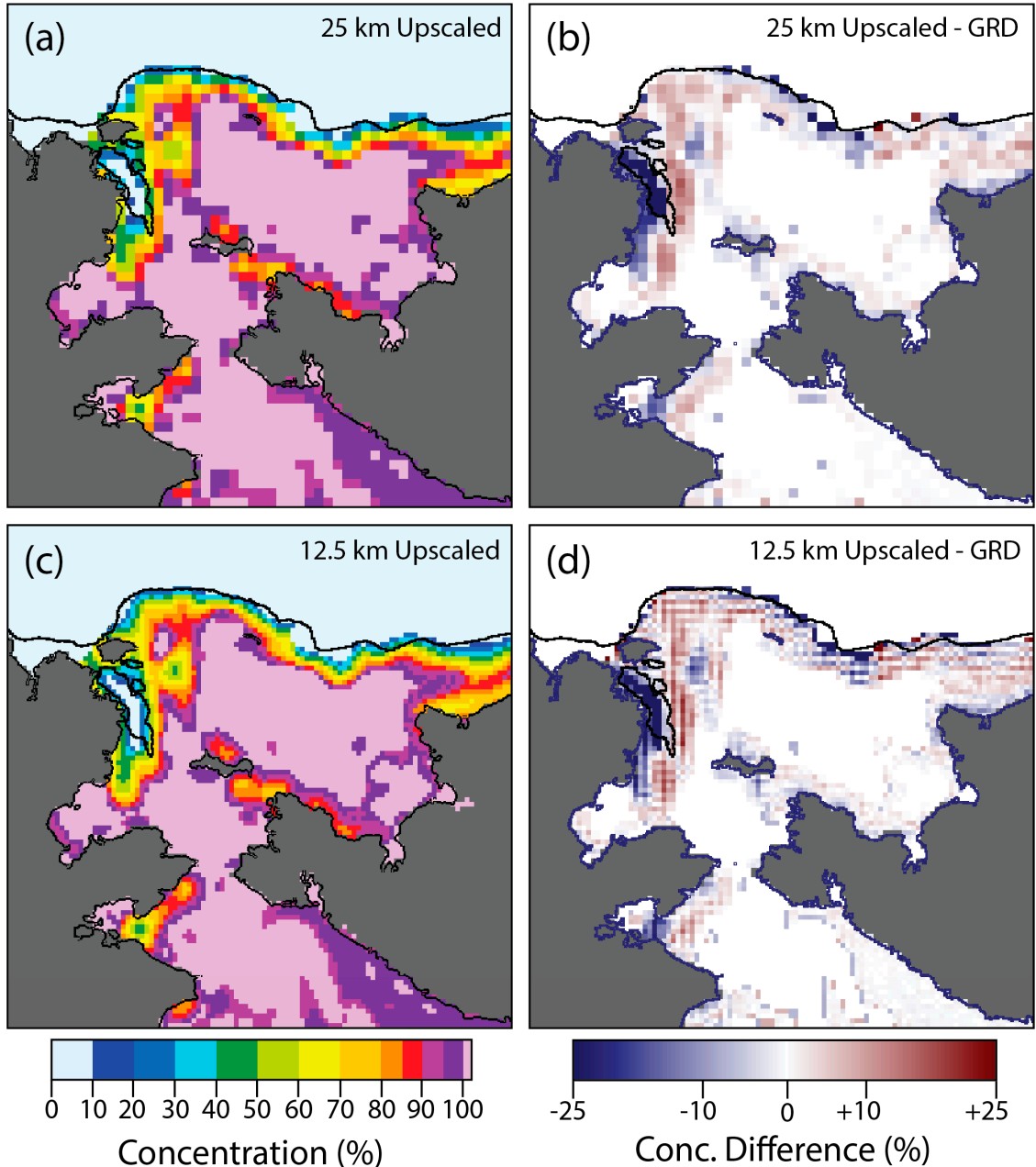

**Figure 6.** Bering Sea 1 March 2014 (**a**) 25 km upscaled rSIR sea ice concentration, (**b**) 25 km upscaled rSIR minus 25 km GRD concentration difference, (**c**) 12.5 km upscaled rSIR concentration, and (**d**) 12.5 km upscaled rSIR minus 25 km GRD concentration difference.

All three show a general consistency with overlaid buoy vectors. We quantitatively assess the PM motions fields by comparing with coincident buoy observations from 1 January through 30 April 2014. For each buoy on each day, the closest PM motion vector was selected for the comparison, with a maximum distance of 50 km from the buoy. This yielded ~2500 buoy-PM pairs for the 25 km GRD and ~7000 for the 12.5 fields. Comparisons were made for the u- and v-components of velocity relative to the EASE2 grid (Table 2). The mean difference (bias) is small for each of the sources, <0.5 cm/s, but the bias in the rSIR motions is about half of the magnitude of the GRD fields. The 12.5 km bi-linear GRD shows some reduction in RMSe, lowering errors by ~0.4 cm/s. The improvement from rSIR is more substantial, reducing the RMSe by more than 1 cm/s (~15%) over the 12.5 km bi-linear GRD and 1.5 cm/s (~20%) over the 25 km GRD.

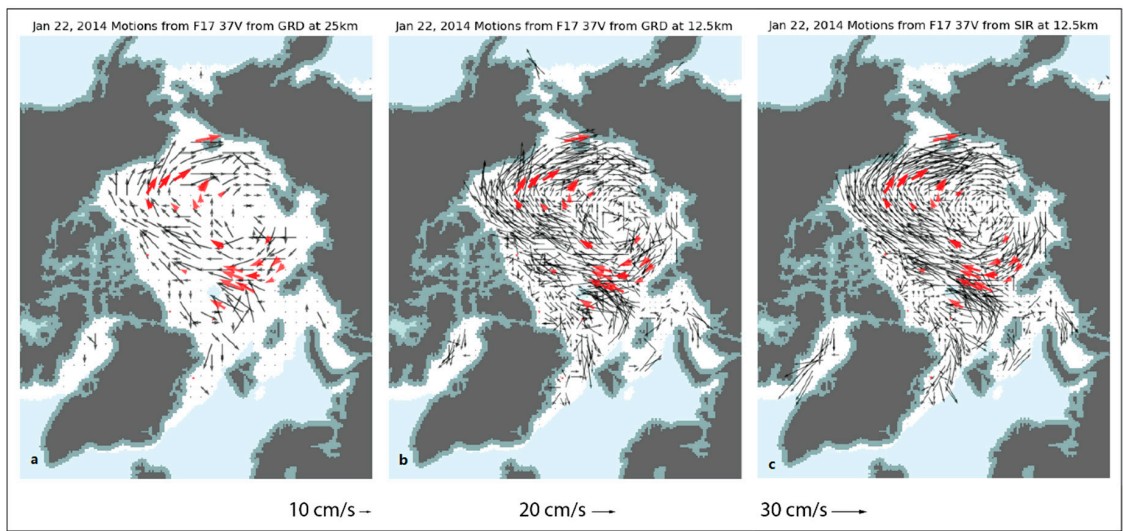

**Figure 7.** Arctic region 22 January 2014 sea ice motion field from 37H TBs for (**a**) 25 km GRD, (**b**) 12.5 km bi-linearly interpolated GRD, and (**c**) 12.5 km upscaled rSIR. Buoy motions are overlaid in red.

**Table 2.** Statistics of PM motions with IABP buoys (PM – buoy) for the standard 25 km GRD resolution, the GRD bi-linearly downscaled to 12.5 km, and the rSIR upscaled to 12.5 km. The nearest PM motion estimate for each buoy was used with a maximum distance of 50 km from the buoy.

| Source | Mean Difference (cm/s) | | RMSe (cm/s) | |
|---|---|---|---|---|
| | u | v | u | v |
| 25 km GRD | −0.22 | 0.25 | 6.55 | 6.41 |
| 12.5 km bi-linear GRD | −0.33 | −0.12 | 6.13 | 6.05 |
| 12.5 km upscale rSIR | −0.13 | −0.05 | 4.99 | 4.99 |

## 4. Discussion

The results here indicate a definitive improvement in the retrieval of passive microwave sea ice parameters using the CETB fields. While the CETB effective resolution is not as high as the gridded resolution provided in the product, upscaling the fields still provides higher resolution (12.5 km vs. 25 km). Furthermore, even at the same resolution, the upscaling appears to be able to capture features in the ice not discriminated by the 25 km GRD fields. For concentration, we have only qualitatively assessed the utility of the CETB, but the potential is clear. The ability to detect more open water regions within the ice (i.e., polynyas) will provide better estimates of heat and moisture fluxes, salinity fluxes, and ice production. The AMSR-E and AMSR2 sensors have higher resolution and have been applied to estimate these parameters [30,31], but the CETB can create a long-term time series of these products, at the same spatial resolution, extending back over 40 years. Furthermore, the CETB includes AMSR-E, which provides the potential for even further improvement in resolution during the AMSR-E and AMSR2 era (AMSR2 is planned to be added when resources allow).

The CETB includes SMMR and all SSMI and SSMIS sensors, so it can be employed to create a long-term sea ice climate record. Historically, the NSIDC DAAC and NASA Goddard have produced one daily sea ice field from a single sensor. Multiple sensors are only used for inter-calibration between sensor transitions. However, for much of the multi-channel passive microwave record, at least two PM sensors have been operating. With the CETB, there is the potential to easily create a gridded ensemble concentration product. From that, extent estimates can be calculated, and long-term extent trends can be calculated, along with an uncertainty estimate based on the variation in extent trends from different sensors.

Another advantage of the CETB is that the TBs from all SSMI and SSMIS sensors have all been intercalibrated and processed consistently. The NSIDC DAAC products have used different TB versions over the years. The sea ice products have been inter-calibrated to adjust for the sensor changes and the Bootstrap product has been reprocessed. However, a consistent full-length TB source with full provenance would provide a stronger foundation for the sea ice climate records.

The twice daily LTOD fields provide yet another avenue for enhancement of sea ice products. While ascending and descending fields are commonly produced (e.g., for AMSR-E and AMSR2), in the polar regions the time of observation between overlapping swaths may differ considerably. The LTOD binning aggregates the TBs into two daily fields that are more consistent in time than ascending/descending binning. Generally, most observations cluster around two 4-h windows, with fewer observations in between [19]. So instead of a 24-h average, effectively an average over a much shorter time is created. This will allow investigation of diurnal changes in the sea ice cover. In some regions, the ice edge can be very dynamic, moving several kilometers in a day; melt and ice growth can also occur on sub-daily time scales. Having twice-daily observations will help discriminate these sub-daily changes. This may be particularly helpful in detecting the onset of melt. There is also a distinct advantage for sea ice motion tracking. As discussed above, motion is calculated by dividing the displacement distance by the time separation. With daily composite TB fields, 24 h is used as the time separation. However, depending on the region, the actual separation can vary between ~2 and nearly 48 h, resulting in large uncertainties in the derived motion fields. In this paper, we used a 24-h separation, but this will be more accurate with the Morning grids because the range in time separation is much smaller than for the daily composite TB fields that have historically been used. Furthermore, with Morning and Evening grids, it is feasible to derive twice-per-day motions and obtain sub-daily motion information.

## 5. Conclusions

Climate records of sea ice parameters have been produced for several decades from a variety of passive microwave brightness temperature sources, sensors, and algorithms. Traditionally, these products have used daily gridded brightness temperatures with drop-in-the-bucket averaging. Here, we explore a new gridded source of TBs from the CETB using the rSIR resolution enhancement method. The results indicate potential enhanced capabilities for sea ice concentration and extent estimates. While the rSIR effective resolution is lower than the product gridded resolution, upscaling the resolution by binning into lower resolution 12.5 km or 25 km fields provides details of the ice cover than are not retrieved by the standard 25 km drop-in-the-bucket GRD fields. Further study is needed to better quantify the improvement in retrievals, but the CETB offers many advantages over previous TB sources.

Sea ice motions are also noticeably improved using the upscaled rSIR fields, decreasing RMS error by 15–20% over the standard GRD product and reducing bias compared to buoy observations. The enhanced resolution retrieves more details in the ice circulation patterns and captures finer gradations in ice motion vectors. The CETB also provides twice-daily fields by LTOD, which yields a more precise time separation and thus more accurate velocities, and there is the potential to obtain sub-daily motion estimates.

Our study here focused on the Arctic, but the CETB also includes fields on an Antarctic grid and in the future, we plan to investigate the applicability in the Antarctic region. Coastal polynyas around Antarctica are important areas of air-sea exchange, ocean energy fluxes, and biological activity. The higher resolution afforded by the CETB may be particularly valuable in assessing long-term changes in these polynyas. The higher resolution will also likely improve accuracy of Antarctic sea ice motion.

We have shown here that the CETB provides useful 19 GHz and 37 GHz TBs at 12.5 km. This is on the same spatial scale as AMSR-E and AMSR2. However, the CETB has a 40+ year record, which has

the potential to track long-term changes in coastal polynyas, fluxes within the ice cover, melt onset and freeze-up, and transport around the Arctic and Antarctic.

**Supplementary Materials:** The following are available online at http://www.mdpi.com/2072-4292/12/16/2552/s1, Figure S1: comparison of land mask, Figure S2: 25 km GRD and 3.125 bi-linearly interpolated GRD, Figure S3: MODIS images zoomed in region of thin ice, Figure S4: land-to-ocean transect for 37 GHz.

**Author Contributions:** Conceptualization, W.N.M. and J.S.S.; methodology, W.N.M. and J.S.S.; software, J.S.S.; formal analysis, J.S.S. and W.N.M.; investigation, J.S.S. and W.N.M.; writing—original draft preparation, W.N.M.; writing—review and editing, W.N.M. and J.S.S.; visualization, J.S.S.; supervision, W.N.M.; project administration, W.N.M.; funding acquisition, W.N.N. All authors have read and agreed to the published version of the manuscript.

**Funding:** This research was funded through the Cooperative Institute for Research in Environmental Sciences (CIRES) Innovative Research Program. Support was also provided by the NASA Earth Science Data Information System (ESDIS) Project through the NASA Snow and Ice Distributed Active Archive Center (DAAC) at NSIDC, grant number 80GSFC18C0102.

**Acknowledgments:** We thank J. Comiso and R. Gersten (NASA Goddard) for providing the Bootstrap algorithm code. We also thank D. Long (BYU) and M.J. Brodzik (NSIDC) for helpful information and guidance on the CETB product.

**Conflicts of Interest:** The authors declare no conflict of interest.

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
