# Peer review of "Assessing the Potential of Enhanced Resolution Gridded Passive Microwave Brightness Temperatures for Retrieval of Sea Ice Parameters"

_remotesensing, doi:10.3390/rs12162552_

Round 1
Reviewer 1 Report
In the manuscript, a new enhanced resolution grid with passive microwave brightness temperature product is used to estimate sea ice concentration and motion, which is modern in approach, because passive microwave data are valuable due to daily coverage of the whole Arctic, are not influenced by clouds, however show significant errors in definition of sea ice parameters. It is a well presented and good writing manuscript, with succinct introduction, clear and detailed results. It gives an image of current research of authors with first results, and it would be interesting to see further study of potential of the Calibrated Enhanced-Resolution Passive Microwave Daily EASE-Grid 2.0 Brightness 101 Temperature (CETB).
Small remarks:
Line 240: the land-ocean, “n” is missed at the end
Line 300 – “motions” is double written
Author Response
In the manuscript, a new enhanced resolution grid with passive microwave brightness temperature product is used to estimate sea ice concentration and motion, which is modern in approach, because passive microwave data are valuable due to daily coverage of the whole Arctic, are not influenced by clouds, however show significant errors in definition of sea ice parameters. It is a well presented and good writing manuscript, with succinct introduction, clear and detailed results. It gives an image of current research of authors with first results, and it would be interesting to see further study of potential of the Calibrated Enhanced-Resolution Passive Microwave Daily EASE-Grid 2.0 Brightness 101 Temperature (CETB).
Thank you for the kind comments.
Small remarks:
Line 240: the land-ocean, “n” is missed at the end
Line 300 – “motions” is double written
Thank you for catching these. Both corrections have been made.
Reviewer 2 Report
3 August 2020
“Assessing the potential of enhanced resolution gridded passive microwave brightness temperatures for retrieval of sea ice parameters” by Meier and Stewart summarizes a new, high resolution sea ice product, and through a case study approach compares it to the traditional 25 km footprint on the (sub-)marginal sea scale (e.g. Bering Sea and Kola Peninsula). The authors also describe some applications of the new enhanced product, including for sea ice motion, which is evaluated and shown to more realistically capture the observed ice movement (from in situ buoys) relative to motion calculations derived from the existing coarse resolution brightness temperatures.
This new dataset, summarized by the paper and shown through the referenced case studies, represents a major contribution to sea ice research. It will be of broad interest to the interdisciplinary Arctic research community not only for updated climatological time series of sea ice concentration and extent, but for applications involving ice age and motion with respect to large-scale atmospheric forcing and anomalous weather events as well as understanding air-sea interactions in fjords and over local communities which is prohibitive at present on the 25 km scale. The manuscript is well-written and organized, and as such I offer just a few minor comments.
Minor Comments:
General comment: As the paper’s proof of concept approach is tilted toward Arctic applications, if there are plans to also create the enhanced resolution product for the Southern Hemisphere it might be worth noting in the Introduction and/or Conclusions some of the advantages for developing this for Antarctic sea ice studies as well.
Line (L) 33: Upon transitioning from SMMR to SSMI sensors the time series became “continuous” at the daily resolution. I see what you mean here regarding the length of the record, but perhaps clarify this point.
L41: What is meant by “…certain conditions” such as? In fjords, coastlines??
L50-51: Instead of “…Imager and Sounded” should it be “Imager Sounder”?
L98: Should this say “…as low as ~2 hours…”?
L141: What is meant by “…a sharp boundary…”? Can you please clarify or be more specific in the description?
L202: Do you mean “…integration of imagery…”?
L289: Would suggest switching from “Northern Hemisphere” to “Arctic Ocean” as some NH sea ice areas (e.g. Hudson Bay) are omitted by the domain choice.
L291-301 (including Table 2 and its caption): Can you briefly provide some additional detail regarding how the differences between PM observations and buoys are calculated in terms of space and time overlap of the observations? Also, remove one of the “motions” on L300.
Author Response
“Assessing the potential of enhanced resolution gridded passive microwave brightness temperatures for retrieval of sea ice parameters” by Meier and Stewart summarizes a new, high resolution sea ice product, and through a case study approach compares it to the traditional 25 km footprint on the (sub-)marginal sea scale (e.g. Bering Sea and Kola Peninsula). The authors also describe some applications of the new enhanced product, including for sea ice motion, which is evaluated and shown to more realistically capture the observed ice movement (from in situ buoys) relative to motion calculations derived from the existing coarse resolution brightness temperatures.
This new dataset, summarized by the paper and shown through the referenced case studies, represents a major contribution to sea ice research. It will be of broad interest to the interdisciplinary Arctic research community not only for updated climatological time series of sea ice concentration and extent, but for applications involving ice age and motion with respect to large-scale atmospheric forcing and anomalous weather events as well as understanding air-sea interactions in fjords and over local communities which is prohibitive at present on the 25 km scale. The manuscript is well-written and organized, and as such I offer just a few minor comments.
Minor Comments:
General comment: As the paper’s proof of concept approach is tilted toward Arctic applications, if there are plans to also create the enhanced resolution product for the Southern Hemisphere it might be worth noting in the Introduction and/or Conclusions some of the advantages for developing this for Antarctic sea ice studies as well.
Thank for this suggestion. While we focused on the Arctic in this paper, the data are available for the Antarctic and same methods should be applicable. We edited the end of the 2nd from last paragraph in Section 2 to mention the Antarctic: “In this paper, we focus on a case study in the Arctic region, but the CETB includes fields for the Antarctic. The goal of this paper is not to conduct a full-scale validation but rather to illustrate the potential of the CETB for improved sea ice fields. In the future, we plan to investigate the CETB further, including examining performance in the Antarctic.”
We also added a paragraph to the conclusion: “Our study here focused on the Arctic, but the CETB includes fields on an Antarctic grid and in the future, we plan to investigate the applicability in the Antarctic. Coastal polynyas around Antarctica are important areas of air-sea exchange, ocean energy fluxes, and biological activity. The higher resolution afforded by the CETB may be particularly valuable in assessing long-term changes in these polynyas. The higher resolution will also likely improve accuracy of Antarctic sea ice motion as well.”
Finally, we changed the last line in the Conclusion to end with “…around the Arctic and Antarctic.”
Line (L) 33: Upon transitioning from SMMR to SSMI sensors the time series became “continuous” at the daily resolution. I see what you mean here regarding the length of the record, but perhaps clarify this point.
We added a sentence to note that the SMMR period, 1978-1987, is every other day.
L41: What is meant by “…certain conditions” such as? In fjords, coastlines??
We added “such as near the ice edge and in coastal regions.” to the sentence.
L50-51: Instead of “…Imager and Sounded” should it be “Imager Sounder”?
We changed (and corrected) it to “Imager/Sounder”, which seems to be the most widely-used convention.
L98: Should this say “…as low as ~2 hours…”?
Yes. This was corrected.
L141: What is meant by “…a sharp boundary…”? Can you please clarify or be more specific in the description?
We added “(where the TB values substantially change in a step-like manner, such as a coastline)”
L202: Do you mean “…integration of imagery…”?
Yes. This was corrected.
L289: Would suggest switching from “Northern Hemisphere” to “Arctic Ocean” as some NH sea ice areas (e.g. Hudson Bay) are omitted by the domain choice.
We changed to “Arctic region” as there are motions in some regions outside of the Arctic Ocean proper – e.g., Baffin Bay, East Greenland Sea, Bering Sea.
L291-301 (including Table 2 and its caption): Can you briefly provide some additional detail regarding how the differences between PM observations and buoys are calculated in terms of space and time overlap of the observations? Also, remove one of the “motions” on L300.
We added to the main text: “For each buoy on each day, the closest PM motion vector was selected for the comparison, with a maximum radius of 50 km from the buoy.” And we added to the Table 2 caption: “The nearest PM motion estimate for each buoy was used with a maximum distance of 50 km from the buoy.”